# Assessing Radish Health during Space Cultivation by Gene Transcription

**DOI:** 10.3390/plants12193458

**Published:** 2023-09-30

**Authors:** Karl H. Hasenstein, Susan P. John, Joshua P. Vandenbrink

**Affiliations:** 1Biology Department, University of Louisiana Lafayette, Lafayette, LA 70504, USA; susan.pratiksha1@louisiana.edu; 2Department of Biological Sciences, Louisiana Tech University, Ruston, LA 71272, USA; jpvdb@latech.edu

**Keywords:** space biology, gene transcription, radish, SPGE

## Abstract

During the Advanced Plant Habitat experiment 2, radish plants were grown in two successive grow-outs on the International Space Station (ISS) for 27 days each. On days 10, 18, and 24, leaf punch (LP) samples were collected and frozen. At harvest, bulb tissue was sampled with oligo-dT functionalized Solid Phase Gene Extraction (SPGE) probes. The space samples were compared with samples from ground controls (GC) grown at the Kennedy Space Center (KSC) under the same conditions as on the ISS, with notably elevated CO_2_ (about 2500 ppm), and from lab plants grown under atmospheric CO_2_ but with light and temperature conditions similar to the KSC control. Genes corresponding to peroxidase (RPP), glucosinolate biosynthesis (GIS), protein binding (CBP), myrosinase (RMA), napin (RSN), and ubiquitin (UBQ) were measured by qPCR. LP from day 24 and bulb samples collected at harvest were compared with RNA-seq data from material that was harvested, frozen, and analyzed after return to Earth. The results showed stable transcription in LP samples in GC but decreasing values in ISS samples during both grow-outs, possibly indicative of stress. SPGE results were similar between GC and ISS samples. However, the RNA-seq analyses showed different transcription profiles than SPGE or LP results, possibly related to localized sampling. RNA-seq of leaf samples showed greater variety than LP data, possibly because of different sampling times. RSN and RPP showed the lowest transcription regardless of method. Temporal analyses showed relatively small changes during plant development in space and in ground controls. This is the first study that compares developmental changes in space-grown plants with ground controls based on a comparison between RNA-seq and qPCR analyses.

## 1. Introduction

Research of plant growth in space has been conducted since the beginning of manned space flight, and the performance of higher plants in microgravity has been obtained from a diverse variety of plants as well as organ, tissue, cell- and protoplast cultures [1,2]. Early investigations established that plants grow successfully in space when physical constraints such as lack of buoyancy are properly mitigated [3] and oxygen [4,5] and nutrient requirements are met [6]. Initially, most space research was conducted on seedlings in Petri dishes in the BRIC (Biological Research In Canisters) hardware because of limited plant cultivation space and scarcity of flight opportunities. The importance of cultivating plants was addressed by developing larger growth facilities such as VEGGIE and the Advanced Plant Habitat, a growth chamber of about 64 L volume [7]. Although still relatively small, these furnishings provide important advances for research [8] and the transition to develop biological life support systems for future plant cultivation in space. The response to reduced gravity [9], altered CO_2_ concentration [10,11], temperature [12], pressure [13,14], and UV [15] and ionizing radiation [16,17,18] typically induce stress-related symptoms in plants. However, higher plants are essential for bioregenerative life support systems (BLSS) because they produce edible biomass and oxygen, recycle water, absorb carbon dioxide, and support the psychological health of astronauts [19,20,21]. Therefore, it is safe to conclude that optimizing plant cultivation will remain an important aspect of future spaceflights and for the establishment of settlements on the Moon and, eventually, Mars.

Despite many reports on plant responses to space conditions, studies that assess developmental changes during space culture are scarce but important because they can identify when adverse effects during space cultivation require intervention before reducing plant performance and productivity. 

This paper evaluates changes in gene transcription during two successive grow outs of radish plants on the International Space Station and compares qPCR analyses with RNA-seq data from tissue collected at harvest. Although physiological evaluations would best be performed in situ, constrained crew time and limited scientific equipment prevent biochemical measurements or quantitative analyses. Therefore, samples were harvested at suitable time intervals, frozen, and analyzed after return to Earth. Because of these constraints, sampling was limited to small sample sizes such as leaf punch disks or Solid Phase Gene Extraction (SPGE) [22,23,24,25] to minimize the effects on plant growth. Rather than analyzing enzyme activities or measuring compounds of interest, this report focuses on the transcription level of relevant genes.

Quantified genes included anabolic and catabolic transcripts for glucosinolates, an important class of metabolites in the Brassicaceae. While typically viewed as defensive compounds against herbivory, glucosinolates contribute to the flavor of Brassicaceae [26], promote human health [27,28,29,30], and serve as indicators of plant health [31]. They are derived from the photorespiratory and glycolytic phosphoserine pathway [32], promoted by jasmonic acid [33], sensitive to auxins [34], and reduced in the presence of ozone [35]. In addition, glucosinolates influence root growth and development [36]. Therefore, glucosinolates represent a wide array of physiological aspects in plants.

The glucosinolate metabolism involves biosynthetic genes (GIS), which modify amino acids, followed by core biosynthesis and side chain modification [37,38]. Cytochrome P450 83A1 is a protein that oxidizes amino acids to aldoximes and initiates glucosinolate biosynthesis [39] and was used to assess glucosinolate anabolism. The activation of vacuole-stored glucosinolates as defense molecules depends on their hydrolysis by myrosinase [40], which is reported as RMA.

Chlorophyll-binding proteins (CBP, AT1G72290, Kunitz family trypsin, and protease inhibitor protein) are sensitive to oxidative stress and are members of a large family of proteins with diverse functions in both light harvesting and photo protection; they stabilize the photosynthetic apparatus and respond to generic stress [41,42]. 

Peroxidase precursors (RPP) are indicators of stress, especially oxidative stress [43]. Damaging reactive oxygen species (ROS) are counteracted by antioxidant systems: low molecular mass antioxidants (ascorbic acid, glutathione, and tocopherols), enzymes regenerating the reduced forms of antioxidants, and ROS-interacting enzymes such as superoxide dismutase, peroxidases, and catalases. Thus, peroxidase precursors serve as indicators for oxidative stress. 

Napin (RSN) is a seed storage protein [44] that is synthesized as a precursor protein at the endoplasmic reticulum and transported along a gradient of decreasing pH to the vacuole, where two pro-peptides are removed to produce the mature protein [45]. Seed storage proteins have antifungal properties and correlate with plant health and productivity [46].

The assessment of these genes provides a diverse framework to detect plant stress. The attempt to investigate these genes by different methodologies aims to identify the most reliable technique for detecting stress. The results showed a strong correlation between qPCR data from ground controls and space experiments. However, the correlation between qPCR and RNA-seq analyses only applied to a subset of the examined genes. Because space experiments are rarely replicated, the reproducibility of separate experiments is difficult to assess and the space syndrome [47] has not been satisfactorily described. This report details the transcription profile of the selected genes that were sampled from leaf material and compared with a ground control and RNA-seq data.

## 2. Results

There was variability between the first and second space experiment, but there were generally consistent profiles over time between the flight and ground control (Figure 1). The transcription values in the LP samples remained low for RSN and RPP and decreased for GIS during the first grow out but increased during the second grow out. CBP showed increased transcription in both flight experiments and ground control over time. RMA showed the lowest transcription at day 18 but recovered toward the end of the experiment in all three experiments. The decrease in ground control was negligible but significant during the first space experiment. The dynamic range for space and ground control data was similar and no consistent differences were noticed. The ground control resembled the second grow-out more than the first grow-out (Figure 2A,B). The consistent decline in transcription of GIS during grow-out 1 (Figure 2A) is indicative of the less robust growth of plants during the first space experiment. The ground control showed transcription profiles that were between the first and second space experiment. Based on all genes, GIS appears to be the most sensitive indicator of plant performance.

### 2.1. Evaluation of qPCR of LP Samples over Time

Comparing the average transcription level for all three sampling times shows that ground control data were comparable to flight data (Figure 2). The flight data showed a similar profile for CBP and RMA but were different between RSN, RPP, and GIS. On average, RMA showed the highest transcription levels, suggesting that RMA either has fewer biosynthetic steps that contribute to myrosinase biosynthesis or that maintaining myrosinase levels requires replenishment and, thus, continuously higher transcription. The latter point can also be deduced from the generally higher level of transcription of RMA during the entire cultivation time (Figure 2 and Figure 3). The comparison of time course data with the average of all flight measurements shows a reduced dynamic range for space-grown plants compared to the ground control (Appendix A).

### 2.2. Comparison between SPGE and RNA-Seq Data of Bulbs

In contrast to repeated leaf measurements, the bulb tissue was sampled only once at the time the growth experiment was terminated. At the time of harvest, the bulb tissue was sampled by functionalized probes (oligo-dT15-coated acupuncture needles [23,48]) that were inserted for one minute into the apical bulb tissue (Appendix A), and then stored at −80 °C in microfuge tubes. The SPGE data (Figure 4A) differ from the values from LP samples (Figure 2 and Figure 3). Most notably, the high transcription of RPP for SPGE contrasts with the low value in LP samples. The intermediate value for GIS in LP samples was replaced by a much-reduced value in SPGE samples. The typically high level for RMA was significantly reduced in SPGE measurements. Comparing ground control with flight data showed higher values in GC for RSN but otherwise consistent transcription patterns. The general consistency between GC and flight samples suggests that SPGE is a suitable method to reliably sample genetic material, even if it must be stored for extended periods. 

Since radish bulbs were also examined by SPGE and RNA-seq, a comparison between these two analyses is appropriate (Figure 4B). Based on FPKM (Fragments Per Kilobase per Million mapped fragments) values, the transcription level for the examined genes was similar for the two flight experiments and the ground controls conducted at the Kennedy Space Center (GC) and university lab (UL), despite the difference in CO_2_ concentrations between the KSC and space (about 2500 ppm) and ambient (400 ppm) in the lab. Although both data sets are based on the average transcription or FPKM values of the five examined genes, SPGE and RNA-seq data show different distributions. The highest transcription by SPGE was observed for RPP for flight experiments and ground control; all other genes were transcribed to a lesser extent. In contrast, the highest transcription level in RNA-seq data was registered for RMA. CBP values were lower based on FPKM values than for qPCR. This observation indicates that qPCR and RNA-seq data do not always correspond with each other, as was also pointed out earlier [49,50]. Nonetheless, FPKM-based values were consistent for all genes. This is likely related to the fact that radish bulbs do not produce seed storage protein (RSN) or photosynthesize and, therefore, do not require peroxidase. Based on this reasoning, the notable transcription of chlorophyll-binding protein (CBP) and RPP in SPGE-sampled, non-photosynthesizing tissue such as bulbs can be explained only by the fact that SPGE samples were taken from outer, apical tissue (Appendix A). 

### 2.3. Comparison between Leaf Punch and RNA-Seq Data

Like the SPGE and RNA-seq assessment, the data set of LP samples from flight and corresponding ground controls (Figure 3 and Figure 4) and the availability of RNA-seq data from leaf material at harvest allows for a comparison of genetic information in leaves (Figure 5) based on different techniques. The LP and RNA-seq data correspond well for RSN and RPP values, but to a lesser extent for higher transcribed genes (GIS, CBP, RMA). While qPCR data were consistent between flight and ground control (Figure 5A), the RNA-seq data showed substantial differences. The differences between KSC and the flight experiment were greater than between lab and flight results (Figure 5B). RNA-seq data for GIS were higher in the KSC extracts but similar for lab and flight. The values for RMA were lower than qPCR but, because of the large error, could not differentiate between flight and either control. Overall, the comparison between qPCR and RNA-seq shows a larger and possibly more sensitive evaluation for the former because the range (−6 to +7, vs. −6 to +2) is about 30 times larger.

### 2.4. Plant Productivity

Although the determination of gene expression changes during growth and its potential to determine the onset of stress is desirable, plant productivity is ultimately assessed by the mass of product, in the case of radishes, the bulb, despite the suitability of leaves for consumption because of their higher nutrient content [6]. A comparison of bulb mass for the two space grow-outs and the ground control showed no statistically significant difference (Figure 6).

## 3. Discussion

The space experiment relied on the cultivation of radishes in the APH and was based on their fast growth rate, small canopy, and limited height. The relatedness to Arabidopsis further allows for a genetic examination of plant responses to space and environmental changes such as elevated CO_2_ concentration. The unique ability to compare two grow-outs to each other and a parallel ground control guided the investigation of plant performance from germination to the harvest of bulbs but, because of time constraints, did not include assessing the transition to generative growth, i.e., bolting and flowering. Because space cultivation has generally been associated with stress [51,52], our investigation aimed at identifying genetic indicators for stress. While all examined genes fall into this category, the genes characterize stress responses either directly (e.g., RPP) or pathways that are likely to be affected by stress-related alteration of physiological parameters. 

While space-grown plants showed changing transcription profiles during growth, none deviated strongly enough from the controls to suggest significant trauma. However, it cannot be ruled out that cultivation in the APH itself induces a certain amount of stress because of constant air movement that is likely to enhance transpiration. Evidence for this notion can be derived from necrotic leaf margins that were observed in space and KSC controls (Appendix A). Thus, at least in radishes and possibly short-lived plants in general, genetic investigations are not needed and observations of turgor and leaf appearance (greenness) are sufficient to maintain successful cultivation in space. However, successful space cultivation relies on optimized environmental conditions. Although the Advanced Plant Habitat has enormous flexibility, the chosen growth conditions may have to be improved.

It is unknown to what extent the collection of leaf punch samples affected the plants, but since excising small leaf samples did not cause specific stress and no observable damage around the perimeter of the excised tissue (Figure 1), it is likely that sampling did not cause measurable damage. Nonetheless, some effect on leaves cannot be ruled out. Limiting the collection of leaf punch samples to three days during the growth cycle is a compromise between potential detrimental effects, desirable temporal resolution, and available crew time. 

The transcription profiles for myrosinase (RMA) during the first and second space experiment (Figure 2) showed a reduction that was not observed in the ground control. This observation suggests that glucosinolate biosynthesis (GIS) and myrosinase (RMA) are linked because as RMA decreased, RSN increased (Figure 2 and Figure 3). 

An interesting corollary to this observation comes from the fact that after the second grow-out, some plants were consumed by the crew. Although a systematic organoleptic analysis was not possible, there was consensus that the flavor of the radishes was identical to radishes that had been consumed on earth prior to space flight. This anecdotal information and the balance between GIS and RMA transcription (Figure 2 and Figure 3) suggest that changes in pungency controlling glucosinolate content were not significant.

In addition to assessing plant health, genetic sampling during space cultivation is also meaningful to assess the possible effect of plant pathogens that typically show greater virulence in space [53]. Although at present, genetic analyses are difficult to perform in space because the required specialized chemicals (primers) and equipment like qPCR systems are either not available or require specific mixtures that cannot be prepared in space. Therefore, leaf punch and SPGE are valuable approaches for tissue sampling. However, attempts to obtain in situ analyses are being developed based on SPGE [24]. Therefore, a comparison between SPGE and tissue samples is relevant (Figure 4). The two sampling techniques showed differences for RPP and RMA, but consistent levels for the other genes. It is possible that the specificity of sampling the apical end of the bulb (Appendix A) rather than the bulk tissue contributes to the discrepancy. The same argument applies to different outcomes between SPGE and RNA-seq analyses (Figure 4). SPGE represents specific tissue, while RNA-seq is rather unspecific and represents bulk tissue. The fact that even after six months of storage, gene transcription could be analyzed from SPGE probes indicates that this sampling method provides a fast and inexpensive resource to examine mRNA from diverse biological materials at a resolution and speed that is unequaled by other macroscopic methods.

The comparison between qPCR from LP and RNA-seq from leaf samples is consistent for low-transcription genes but does not resolve elevated values as well as qPCR data. Although the difference could be the result of the difference between 24 days for qPCR and 27 days for RNA-seq, the comparison is unlikely to be related to obvious changes like senescence or chlorosis occurring during the last days of cultivation and any changes in gene expression as a result of the transition to generative growth, i.e., flower formation, typically occurs after more than 50 days [54,55]. The difference between the techniques is substantial for genes that are transcribed higher than the average, especially for RMA (Figure 5). This observation is inconsistent with relying on RNA-seq to determine suitable reference genes for qPCR [56]. In addition, qPCR analyses were more consistent between the ground control and flight experiment, further supporting the validity of qPCR analyses. However, the precise quantification of gene transcription by either qPCR or RNA-seq remains a challenge that is likely amplified by the sensitivity of individual organisms and biochemical and spatial constraints such as gene loss and RNA arrangements that affect sequencing and read depth [57].

Studies of plant growth in space identified different types of stress that collectively are referred to as space syndrome [47], a combined effect of space and environmental factors that affect plant growth. The goal of providing engineering solutions, such as fan-induced air movement to counteract the lack of buoyancy, may overcompensate for the original problem and induce new stresses (Appendix A). The commonly reported elevation of stress-indicating genes in space experiments [51,52] has not been examined in this study and may not have been detected. However, the indistinguishable productivity between the first and second space experiment and the ground control (Figure 6) suggests that plant productivity in space was not significantly diminished. Either the postulated space stress was too low to affect productivity or the natural variability of plant productivity (bulb formation) masked space effects. The latter issue can only be resolved with larger sample sizes.

The observed increase in gene transcription at the last sampling time (days 24 and 27) may be the result of maturation or aging, rather than stress. Therefore, this study provides evidence that space cultivation of crops can provide fresh and nutritious food that can be grown in a limited space at high spatial density (five radish plants in a science carrier quadrant or 20 by 20 cm). Although there were transcriptional differences between the two grow-outs, it is unclear if they were significant or represent natural variability among plants, regardless of culture conditions. To our knowledge, no comparable studies exist on earth-cultivated plants.

In summary, the examination of gene transcription does not show obvious stress responses. However, it is important to remember that although transcription data are often mistaken for protein expression data, post-translational modifications are important for biological control, but they have not been examined in this study. Transcriptomic studies should ideally be accompanied by comprehensive metabolomic investigations to corroborate the transcriptome information.

## 4. Methods

We used *Raphanus sativus* variety “Cherry Belle” seeds for the experiments. The seeds were sanitized and grown in arcillite that was fertilized in half-strength MS medium without chloride and grown as described previously [6]. The space experiment was launched 2 Octorber 2020 and two grow-outs were performed in November and December 2020 in the Advanced Plant Habitat (APH) and returned in July 2021. Ground controls were performed in December 2020 at the Kennedy Space Center and in May 2021 at the university laboratory. 

Leaf punch samples (6 mm diameter leaf disks) were collected 10, 18, and 25 days after imbibition and stored in microfuge tubes until sample return and subsequent analyses. SPGE probes were prepared as described earlier [48] and stored on the ISS for 65 days prior to use at ambient temperature. SPGE sampling of radish bulbs was performed at the apical end of radish bulbs 27 days after imbibition. LP samples and SPGE probes were stored at −80° C for 6 months before return to Earth. RNA from leaf punch data was extracted with Spectrum™ Plant Total RNA Kit (Sigma, St. Louis, MO, USA) from samples, which were macerated in liquid nitrogen. RNA from SPGE probes was released by 10 min incubation in 10 μL of 80° C water. All samples were reverse transcribed using SuperScript^®^ III Reverse Transcriptase (Life Technologies, Carlsbad, CA, USA). The chosen transcripts were analyzed using q-PCR on a Step-One Real-Time PCR system (Applied Biosystems, Waltham, MA, USA) using primers listed in Appendix A, as described previously [48]. Transcription was expressed relative to the average of all examined genes for SPGE, qPCR, and RNA-seq data.

RNA-seq analysis was performed from frozen tissue that was extracted in liquid N_2_, quality checked and processed by Novogene Corporation, Sacramento, CA, USA. FPKM of genes corresponding to the qPCR primer sequences were transformed [(FPKM + 1)/average FPKM_n_)] before log(2) conversion.

## 5. Conclusions

Leaf punch, SPGE and RNA-seq data show that the transcription of genes varies during the cultivation of radish plants between space and ground controls. Monitoring the variability of transcriptions is a sensitive way to monitor plant performance but did not reveal major stress responses. Future cultivation of plants on the Moon or Mars is required to support human settlements and transcriptional assessment will detect plant stress and avoid crop loss.

## Figures and Tables

**Figure 1 plants-12-03458-f001:**
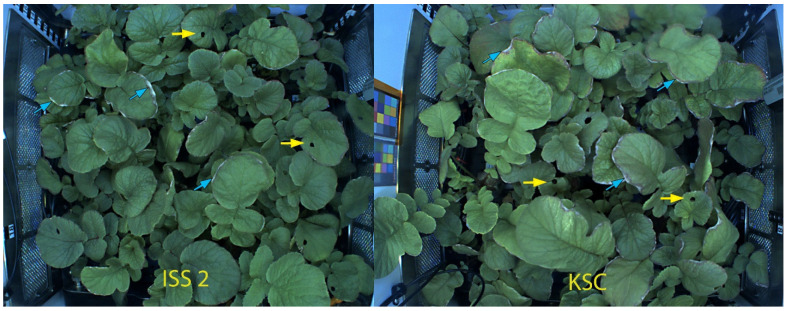
Top view of radish plants grown in the Advanced Plant Habitat on the last day of the second space experiment (ISS) and the ground control (KSC). Yellow arrows indicate leaf punch sampling sites, and the blue arrows indicate necrotic margins of leaves. Leaf punch holes are six mm in diameter at the time of sampling.

**Figure 2 plants-12-03458-f002:**
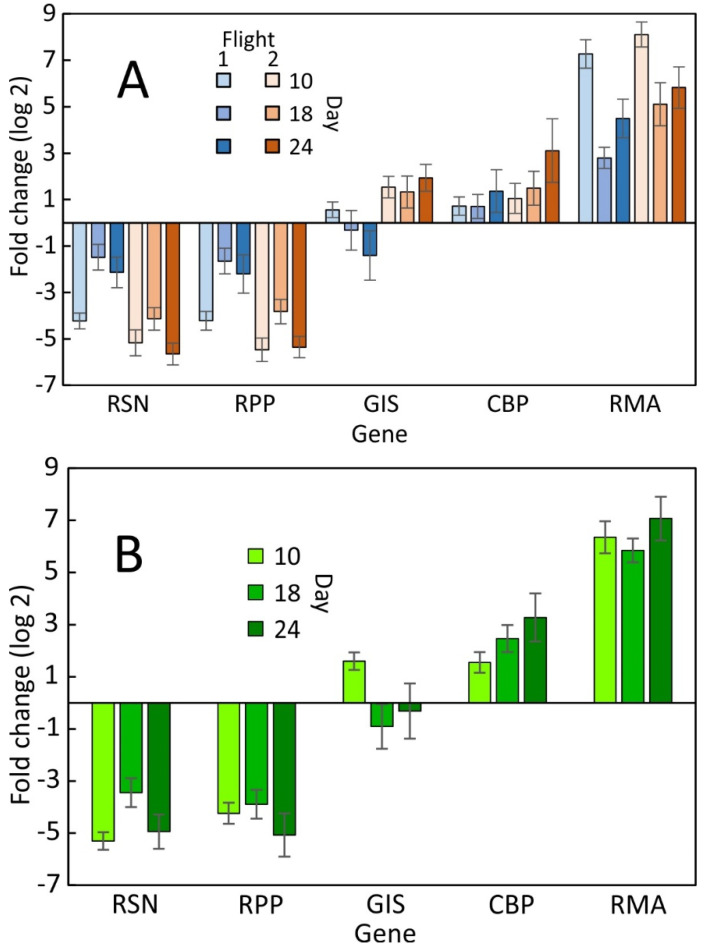
Relative expression of five genes during cultivation of *R. sativus* in space aboard the International Space station (**A**) in comparison with a ground control experiment at the Kennedy Space Center (**B**). The data are the result of eight independent measurements with three technical replicates each relative to the average of all genes obtained from leaf punch samples that were collected on day 10, 18, and 24 of a 27-day cultivation period. Average ± SE.

**Figure 3 plants-12-03458-f003:**
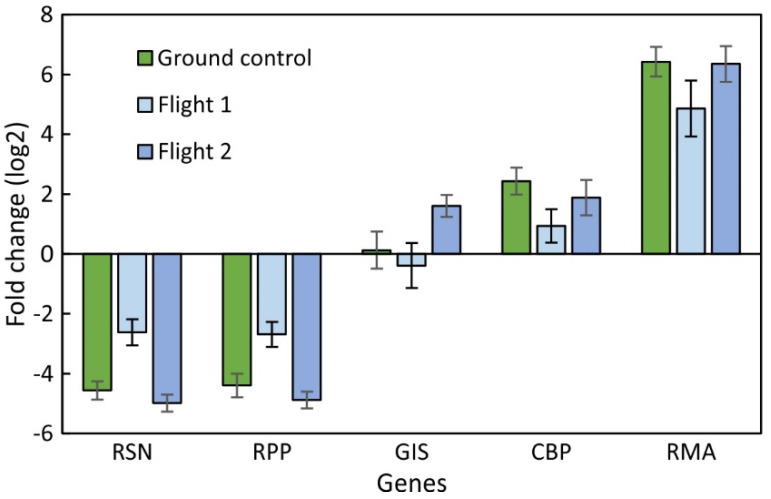
Average transcription levels of selected genes in ground controls conducted at the Kennedy Space Center and during two successive space cultivations (Flight 1 and Flight 2) of *R. sativus*. The data represent pooled leaf punch data collected on day 10, 18, and 24 of a 27-day cultivation period relative to the average of all genes. Average ± SE.

**Figure 4 plants-12-03458-f004:**
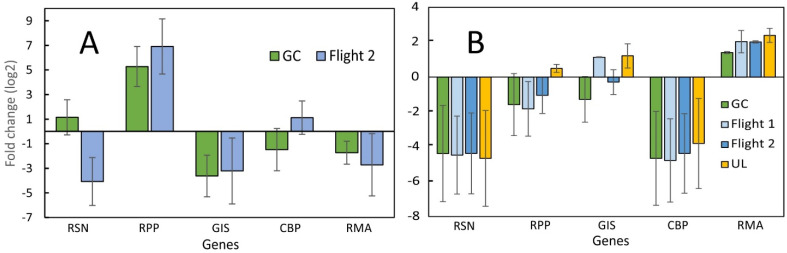
(**A**) SPGE analysis of transcription levels of selected genes of *R. sativus* collected from the second space cultivation (Flight) in comparison with a ground control experiment (GC) conducted at the Kennedy Space Center. (**B**) radish bulb RNA-seq data for the same genes obtained from the first and second space experiment (Flight), a ground control with the same elevated CO_2_ concentration as on the ISS (Ground control), and a grow-out at ambient CO_2_ (UL). The data represent the average of triplicate SPGE samples, and four RNA-seq replicates collected 27 days after imbibition. Radishes and SPGE probes were frozen, stored in space for six months, and processed after return to Earth or processed within a week (Lab). The data are shown as fold-changes relative to the average of all selected genes for each experiment; average ± SE.

**Figure 5 plants-12-03458-f005:**
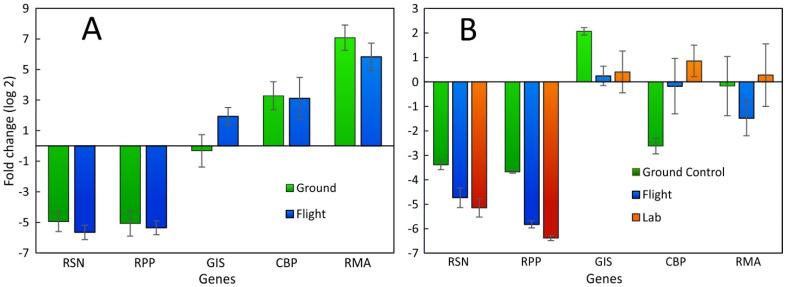
(**A**) Transcription levels of selected genes from radish LP samples at 24 days from the ground control and the second flight experiment (a subset of Figure 1). (**B**) Gene transcription data of the same genes based on FPKM values of RNA-seq analysis of radish leaves collected at harvest (day 27). The data represent the average of triplicate LP samples and four RNA-seq replicates. All values are expressed as fold-changes relative to the average for each set of genes; averages ± SE.

**Figure 6 plants-12-03458-f006:**
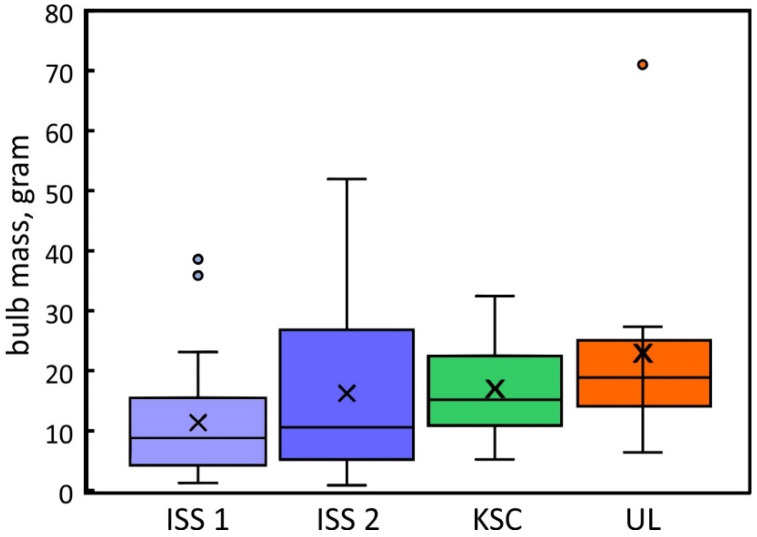
The mass distribution of radish bulbs grown in space (ISS1, ISS2) and corresponding ground controls (KSC and UL). All experiments were performed in the Advanced Plant Habitat at about 2500 ppm CO_2_. The means (X) and averages (horizontal lines) were not statistically different (*p* = 0.32). Distributions are based on the weight of 20 (GC, ISS1) and 10 bulbs (ISS2, UL).

## Data Availability

Data are available upon request from the corresponding author.

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
