# Peer review of "Assessing Radish Health during Space Cultivation by Gene Transcription"

_plants, 2023, doi:10.3390/plants12193458_

Round 1

Reviewer 1 Report

The manuscript investigated the growth and gene expression of radish during space cultivation. The results showed that the transcription of genes varies during the cultivation of radish plants between space and ground controls. The study provided novel information of plant health during space cultivation.

Some minor comments:

Line 3, delete “.”

Line 99-103, Move these sentences to the introduction.

Line 119, add scale bar

Author Response

Thanks for your review. I have complied with all you recommendations and moved the first paragraph of the results to the introduction, deleted the superfluous period after the title, and added information to Figure one. However, instead of a scale bar (which requires consistent dimensions) I added the diameter of the leaf holes to the figure caption. This scaling provides variable (by distance and angle) but correct information to the entire image.

Reviewer 2 Report

The paper of Hasenstein et al. (registered under the ID# 2600128) deals with a comparative study performed on Radish plants cultivated during two successive flights and the ground control. The author used in their experiments R. Sativus variety “Cherry Belle”. In their investigations, authors focused on leaf punch samples at different growth stages 10, 18 and 24 days and the bulbs at the harvest (27 days). They carried out their different experiment sets on 27-day old plants. Authors to evaluate the Radish heath during the space cultivation in comparison to the corresponding ground controls; they concentrated mainly on the qRT-PCR and RNA-Seq data to assess the genes transcription level abundance. Authors showed that the transcription of 5 crucial genes varies during the cultivation of radish plants between space and ground controls. These changes in genes transcription are, very likely, associated with the maturation or aging processes, rather than stress. Authors mostly performed a descriptive study and no robust conclusions were drawn, but still remains valuable work, where authors exerted a lot of efforts and exhaustive work to coordinate their work between ground and space for the benefits of astronauts crew, in the future. We appreciate their efforts to cultivate and compare plants grown in space and other on the ground. This work can be considered for publication, if my colleague(s) agree(s) with me

- Line 23: you mentioned “RSP” gene but you did not say to what refers the “RSP” gene. Please verify this point. Example “RNS” refers to napin, “RMA” refers to myrosinase; but RSP you did not mention that.

Author Response

Thank you for catching a typo. RSP was corrected to RPP. 

Reviewer 3 Report

Hasenstein and his colleagues aim to assess radish health during space cultivation by comparing gene expression between space-grown plants and ground controls. The manuscript enriches our knowledge about the space-grown plants. The conclusions are supported by the data, and the submitted manuscript is written clearly and general interest to the readers. However, I have several comments that should be addressed before publication.

In scientific aspects, I have some comments:

1.       Why CBP values were lower based on FPKM values than for qPCR, and qPCR and RNA-seq data do not always correspond with each other” (Page5, L162-163). The authors should further discuss in the discussion part.

2.       Why Plant productivity show no statistically significant difference between two space grow outs and the ground control. The authors should further discuss in the discussion part.

Hasenstein and his colleagues aim to assess radish health during space cultivation by comparing gene expression between space-grown plants and ground controls. The manuscript enriches our knowledge about the space-grown plants. The conclusions are supported by the data, and the submitted manuscript is written clearly and general interest to the readers. However, I have several comments that should be addressed before publication.

In scientific aspects, I have some comments:

1.       Why CBP values were lower based on FPKM values than for qPCR, and qPCR and RNA-seq data do not always correspond with each other” (Page5, L162-163). The authors should further discuss in the discussion part.

2.       Why Plant productivity show no statistically significant difference between two space grow outs and the ground control. The authors should further discuss in the discussion part.

In language aspect:

1.  “Show” should be instead of “Showed”, or “were” should be instead of “are” (Page3, L124-125).

2.  “-80C” should be replaced by “-80℃” (Page4, L137-138), Please check.

3.  The authors must carefully check grammar, punctuation, spelling, and overall style in the whole text.

Author Response

We corrected the tense errors and changed are to were and show to showed (L 131). We further corrected 80 C to 80°C (L145).

The request to further discuss the discrepancy between qPCR and RNA seq in the discussion was addressed by inserting the following sentence: (L292): 

However, the precise quantification of gene transcription by either qPCR or RNA-seq remains a challenge that is likely amplified by the sensitivity of individual organisms and biochemical and spatial constraints such as gene loss and RNA arrangements that affect sequencing and read depth [57].

The request to discuss the lack of statitical difference between the space grow-outs was addressed by the following sentence (inserted in L 304): 

Either the postulated space stress was too low to affect productivity or the natural variability of plant productivity (bulb formation) masked space effects. The latter issue can only be resolved with larger sample sizes.